# Role of Endoscopic Ultrasound in the Evaluation of Pancreatic Cystic Neoplasms: A Concise Review

**DOI:** 10.3390/diagnostics13040705

**Published:** 2023-02-13

**Authors:** Shiva Rangwani, Wasseem Juakiem, Somashekar G. Krishna, Samer El-Dika

**Affiliations:** 1Department of Internal Medicine, Ohio State University College of Medicine, Columbus, OH 43210, USA; 2Department of Internal Medicine, Stanford University, Stanford, CA 94305, USA; 3Division of Gastroenterology and Hepatology, Stanford University, Stanford, CA 94305, USA; 4Department of Gastroenterology, Hepatology, and Nutrition, Ohio State University Wexner Medical Center, Columbus, OH 43210, USA

**Keywords:** endoscopic ultrasound, pancreatic cystic lesions, pancreatic cystic neoplasms, fine needle acquisition, intraductal papillary mucinous neoplasm, serous cystadenoma, mucinous cystic neoplasm

## Abstract

Pancreatic cystic lesions are being discovered as incidental lesions during cross-sectional imaging studies of the abdomen with increasing frequency. Endoscopic ultrasound is an important diagnostic modality for managing pancreatic cystic lesions. There are various types of pancreatic cystic lesions, from benign to malignant. Endoscopic ultrasound has a multifactorial role in delineating the morphology of pancreatic cystic lesions, ranging from fluid and tissue acquisition for analysis—fine needle aspiration and through-the-needle biopsy, respectively—to advanced imaging techniques, such as contrast-harmonic mode endoscopic ultrasound and EUS-guided needle-based confocal laser endomicroscopy. In this review, we will summarize and provide an update on the specific role of EUS in the management of pancreatic cystic lesions.

## 1. Introduction

With the abundant use of cross-sectional imaging in medical practice, pancreatic cystic lesions (PCLs) are detected in 13.5–49.1% of adult patients [1]. When encountering a PCL, clinicians need to diagnose and determine its malignant potential. Defining a PCL’s malignant potential is paramount, as it will determine the next course of action, which can range from reassurance to surveillance to surgical resection. As there is a multitude of PCLs with varied morphologies, accurate risk stratification of these lesions has been difficult. Despite multiple guidelines, risk stratification falls short: 25–40% of patients undergoing resection have PCLs with no malignant potential, and 50–75% of mucin-producing cysts resected do not reveal high-grade dysplasia/adenocarcinoma at the final histopathological analysis [2,3]. Since cross-sectional imaging alone has limited sensitivity and specificity in evaluating PCLs with respect to their nature and malignant potential, endoscopic ultrasound (EUS) with and without tissue–fluid acquisition (TFA) is often used for comprehensive management. This review will highlight the indications for performing EUS, the evaluation of PCLs, and the accuracy of EUS imaging modalities with and without TFA in delineating the nature of the PCL of interest.

## 2. Indications to Perform EUS in the Evaluation of PCLs

Indications for EUS evaluation vary in clinical practice, reflecting differing recommendations in many current guidelines, as shown in Table 1 [4,5,6,7]. Each guideline in Table 1 uses imaging characteristics to risk stratify PCLs based on cyst morphology, cyst size, diameter of the main pancreatic duct (MPD), cyst growth rate, and presence of a mural nodule. They also rely on certain clinical characteristics and standard of care testing to differentiate between PCLs with low and high malignant potential. Although they have similarities, there are subtle differences among the guidelines that would trigger an EUS evaluation. For example, the American Gastroenterological Association (AGA), the International Consensus Guidelines (ICG), and the American College of Gastroenterology (ACG) guidelines recommend EUS evaluation for a PCL diameter greater than 3 cm, while the European guidelines recommend EUS if the PCL diameter is greater than or equal to 4 cm. Moreover, cyst growth rates differ between guidelines—ICG recommends evaluation for cyst growth of >5 mm over 2 years, while the ACG guidelines cutoff is >3 mm/year. Additionally, the four guidelines have MPD dilation as a criterion for EUS evaluation; however, only the ICG and European guidelines quantify the dilation deemed appropriate for referral.

Furthermore, there is variation among guidelines in profiling a patient in terms of risk stratification of PCL and predicting the risk of malignancy. For example, ICG describes lymphadenopathy as a feature to order EUS, while the ACG guidelines suggest EUS evaluation for patients with obstructive jaundice or pancreatitis secondary to PCL.

## 3. EUS Imaging of PCLs

The goal of EUS imaging is to help reveal the nature of the PCL and evaluate it for malignancy or the potential for malignant transformation. EUS imaging helps delineate cyst size and allows clear visualization of the cyst wall while evaluating for septations, calcifications, or mural nodules. It also shows the PCL in relation to the surrounding organs and vasculature. In addition, it allows the measurement of the MPD diameter and evaluates the PCL for communication with the duct or its side branches.

### 3.1. EUS Imaging and Morphology in the Differentiation of PCLs

#### 3.1.1. Serous Cystadenomas (SCN)

Serous cystadenomas are commonly found in middle-aged women, with CT or MRI imaging showing either a microcystic or macrocystic appearance [8]. These PCLs have a low malignant potential of around 0.01% [8]. A microcystic and honeycomb appearance of the PCL on EUS imaging is diagnostic of serous cystadenoma (SCN), which can be seen in Figure 1 [9]. This appearance stems from the coalescence of multiple cysts millimeters in diameter. A unilocular SCN can be seen in Figure 2D. In SCNs, the cystic spaces can be separated by fibrous septa that can coalesce into a central scar that may have calcifications, which can be seen in the cytology in Figure 3. However, a smaller percentage of SCNs can be oligocystic, macrocystic, or rarely unilocular, with an EUS appearance that overlaps with other types of PCLs, rendering it challenging to differentiate them [10,11,12].

#### 3.1.2. Intraductal Papillary Mucinous Neoplasm

Branch duct (BD) IPMNs show clear communication with the MPD or its branches, generally in the absence of other EUS changes suggestive of chronic pancreatitis. IPMNs are morphologically classified relative to the extent of involvement of the ductal system as main duct (MD), BD, and mixed type (MT). MD-IPMN results in an abrupt dilation of the main pancreatic duct and can be associated with a “fish-eye” ampulla extruding mucin seen during endoscopic examination, which is pathognomonic for MD-IPMN [13]. MD-IPMNs have variable malignant potential ranging from 38–68% [13]. BD-IPMN is the result of the dilation of side branches of the main pancreatic duct, resulting in a “grape-like” cystic lesion. These represent an intermediate malignancy potential ranging from 15–17%; however, without EUS evaluation, a multitude of these lesions are inappropriately resected [6,14]. A multicenter study reported that 63% of resected BD-IPMNs have low histopathological concern for malignancy [15,16]. MT-IPMN has features of both MD-IPMN and SB-IPMN. Another distinguishing characteristic of IPMNs is that they are multifocal in 5–10% of cases [17]. Histopathologic specimens of low- and high-grade IPMNs are shown in Figure 4. Mural nodularity can be seen in EUS imaging of IPMNs, as shown in Figure 2A.

#### 3.1.3. Mucinous Cystic Neoplasms (MCNs)

MCNs are seen mostly in the body and tail of the pancreas and are almost exclusively found in women. They are mostly unilocular or septated macrocystic cysts that do not communicate with the MPD [18,19,20]. The endosonographic appearance of a thin-walled MCN is shown in Figure 2B. The malignant potential of MCNs is cited at approximately 10% [21].

#### 3.1.4. Pancreatic Neuroendocrine Tumor (pNET) and Solid Pseudopapillary Tumor (SPT)

Pancreatic neuroendocrine tumors (pNETs) and solid pseudopapillary tumors (SPTs) can have cystic components in 8.02% and 33.33% of cases, respectively [22]. PNETs are usually more homogenous (52.80% vs. 14.29%) and hypervascular (62.91% vs. 19.05%) compared to SPT, while calcifications can be observed in 38.1% of SPTs and 4.32% of pNETs [22]. The malignant potential of pNETs ranges from 6–31%, while that of SPT is cited at 10% [23,24].

#### 3.1.5. Pancreatic Pseudocysts

Pancreatic pseudocysts arise across a spectrum of clinical situations but are most commonly found in the setting of pancreatitis, with an incidence of 5–16% in acute pancreatitis and 20–40% in chronic pancreatitis [25]. Pancreatitis causes disruption of the main pancreatic duct or its branches, leading to extravasation of pancreatic enzymes and the eventual formation of a distinct collection. Findings suggestive of a pancreatic pseudocyst during EUS evaluation are fluid collections with intracystic debris (as seen in Figure 2C) but without internal septations or mural nodules. Endosonography of the pancreatic parenchyma may reveal evidence of chronic pancreatitis, such as lobularity, side branch ectasia, or calcifications (hyperechoic foci) [26].

#### 3.1.6. Differentiation of PCLs Based on EUS Morphology

The accuracy of EUS morphology in the differentiation of PCLs varies among different studies. In one of the earlier studies, the accuracy of EUS for differentiating PCLs was reported in 52 resected specimens to be 92–96% [27]. On the other hand, Brugge et al. showed that EUS morphological criteria have a limited sensitivity, specificity, and accuracy for characterizing different types of pancreatic cysts (56.1%, 45.4%, and 50.9%, respectively) [28]. Furthermore, despite the contrast and spatial resolution imaging provided by EUS, it is an operator-dependent modality with poor to fair interobserver agreement in differentiating different types of PCLs, with kappa values ranging from 0.16 to 0.53 [29,30].

### 3.2. EUS Imaging and Detection of Malignancy in Mucinous PCLs

Features concerning for malignancy during the evaluation of PCLs are large cyst size, thick septations, thick wall, cyst causing bile duct obstruction, dilated MPD, and presence of a mural nodule [6]. A meta-analysis that included 41 studies evaluated the risk of malignancy in IPMN and reported that size > 3 cm (OR: 62.4; 95% CI: 30.8–126.3), presence of mural nodules (OR: 9.3; 95% CI: 5.3–16.1), dilatation of the MPD (OR: 7.27; 95% CI: 3.0–17.4), and main vs. branch duct IPMN (OR: 4.7; 95% CI: 3.3–6.9) are independent risk factors for malignant transformation [31]. Another meta-analysis showed that the size of the mural nodule is a significant predictor of invasive carcinoma or high-grade dysplasia in IPMNs; however, due to the heterogeneity of the data, a cutoff size was not estimated [32]. Other studies reporting an optimal cutoff in the size of mural nodules measured by EUS for predicting malignant IPMNs are 5, 7, and 10 mm [33,34,35,36]. In a 2019 study, Iwaya et al. evaluated the optimal cutoff size of septal thickness using EUS in IPMNs, and they found that septal thickness was an independent predictive factor for malignancy [37]. A septal thickness cutoff value of 2.5 mm provides an accurate prediction of malignant IPMN, with an odds ratio of 3.51 (*p* = 0.003). The area under the curve for the diagnosis of malignancy using septal thickness was 0.74, with better performance of EUS compared to CT (0.70 for EUS and 0.56 for CT). Multivariate analysis showed that the odds ratio for septal thickness ≥ 2.5 mm was 3.51 (95% CI: 1.55–7.97, *p* = 0.003).

A critical issue that endosonographers face in the evaluation of mural nodules within PCLs is the misperception of mucous globules for mural nodules. One way to distinguish between the two is echogenicity. Mucus globules are hypoechoic, with a hyperechoic rim and smooth surface. On the other hand, mural nodules are iso- or hyperechoic with irregular margins. However, the modest accuracy of fundamental B-mode EUS (FB-EUS) in depicting the nature and malignant potential of PCLs, especially when it comes to distinguishing mural nodules from mucous globules, has pushed endosonographers to explore contrast-harmonic mode EUS (CH-EUS) in the evaluation of these lesions.

#### 3.2.1. Advances in Endosonographic Imaging for Defining Cyst Morphology (Diagnosis and Risk Stratification)

##### Contrast-Harmonic Mode Endoscopic Ultrasound

CH-EUS allows for enhanced evaluation and requires intravenous injection of a contrast agent. Within 25–30 s from injection, different patterns of enhancement can be observed relative to the surrounding tissue: nonenhancement, hypoenhancement, and hyperenhancement [38]. The literature does not report significant benefits of CH-EUS over FB-EUS with respect to the characterization of PCLs into different types; however, the role of CH-EUS comes into play when evaluating mural nodules [38]. European guidelines recommend CH-EUS for further evaluation of mural nodules, as it helps to better assess the vascularity within the nodule and identify hyperenhancement concerning for malignancy [19]. CH-EUS has been shown to differentiate unenhanced mucus globules or debris from mural nodules with sensitivity, specificity, positive predictive value, negative predictive value, and accuracy of 100%, 80%, 92%, 100%, and 94%, respectively [39]. One meta-analysis included eight studies (320 PCLs) using dedicated CH-EUS for the diagnosis of mural nodules harboring high-grade dysplasia or invasive carcinoma [40]. Five of these studies (220 PCLs) had surgical pathology as the reference standard, while the other three had surgery, EUS tissue acquisition (EUS-TA), or clinical follow-up as their reference standards. The pooled sensitivity was noted to be 97.0%, and the pooled specificity was 90.4%. A recent study supported the conclusion of the meta-analysis by showing that CH-EUS fine needle aspiration (FNA) through an enhanced mural nodule in PCLs was positive for dysplasia/malignancy in 100% of cases and high-grade dysplasia or malignancy in 76.9% of cases [38]. What was noticeable in this study is the fact that most of these nodules had no signal on FB-EUS.

##### EUS-Guided Needle-Based Confocal Laser Endomicroscopy (EUS-nCLE)

EUS-nCLE utilizes a laser beam to evaluate microscopic images of cyst wall epithelium in vivo. In EUS-nCLE, the endoscopist injects a contrast solution consisting of fluorescein sodium intravenously and advances a probe for image acquisition into the PCL after obtaining cystic access via a 19-gauge needle. EUS-nCLE allows for real-time, high-resolution microscopic imaging of tissue, thereby facilitating in vivo histopathology. Each PCL epithelium has a specific morphology. For example, IPMNs are characterized by finger-line papillary projections with an inner vascular core, MCNs by horizontal-type epithelial bands of variable thickness without papillary conformation, and SCNs by superficial vascular networks [41,42]. EUS-nCLE accurately differentiates mucinous PCLs, with a sensitivity of 98%, specificity of 94%, and accuracy of 97% [41]. Additionally, a recent meta-analysis showed that EUS-nCLE provided a higher pooled diagnostic when compared to EUS-through-the-needle biopsy (EUS-TTNB) (85%, 95% CI: 82–88% vs. 74%, 95% CI: 69–78%, *p* < 0.001) [42]. The same meta-analysis showed similar adverse event rates among the aforementioned evaluative modalities.

## 4. EUS-Fluid Acquisition

A holistic evaluation of a PCL includes an analysis of the cystic fluid. There have been numerous attempts at identifying an ideal biomarker that purports malignant potential for PCLs; however, cyst fluid analysis alone has a low diagnostic yield of approximately 50% [43]. When used in conjunction with EUS and advanced EUS-based imaging techniques, cyst fluid aspiration analysis aims to augment the diagnostic capacity of the evaluating physician. Routine evaluation of PCL fluid includes carcinoembryonic antigen (CEA) and amylase levels, as well as cyst fluid cytology. CEA does help differentiate between mucinous and non-mucinous lesions with varying accuracy depending on cutoff values. In a multicenter prospective study by Brugge et al., a cutoff CEA value of 192 ng/mL had sensitivity, specificity, and accuracy of 75%, 84%, and 79%, respectively [28]. Using a lower cutoff of 105 ng/mL yielded lower sensitivity (70%) and specificity (63%) [44]. Although an agreed-upon CEA upper limit cutoff value is elusive in terms of differentiating mucinous vs. non-mucinous PCLs, it is generally agreed upon that CEA < 5 ng/mL is highly specific for non-mucinous PCL, with a specificity reaching 95%, and CEA > 800 ng/mL is highly specific for mucinous PCL, with a specificity of 98% [45,46,47]. Therefore, clinicians can utilize CEA levels to assess the probability of a PCL being mucinous. Newer PCL fluid markers are being evaluated and analyzed to aid in differentiation: a recent meta-analysis showed that cyst glucose concentration is more predictive of mucinous lesions than CEA, with a cyst fluid glucose of <50 mg/dL indicating a mucinous cyst with a sensitivity of 91% and specificity of 75% vs. 67% and 80%, respectively, for CEA in the same study population [48].

Although the aforementioned cyst fluid tests provide diagnostic information to clinicians evaluating PCLs, they have poor diagnostic yield, fall short in the sensitivity and specificity of determining malignant potential, and are also unable to differentiate between PCL subtypes. Newer testing methods analyze PCL fluid for targeted DNA mutations in order to risk stratify and differentiate PCLs, which is known as next-generation sequencing (NGS). Mutations such as mitogen-associated protein kinase (MAPK) genes (KRAS and BRAF) and guanine nucleotide binding protein–alpha subunit (GNAS) are specific to mucinous PCLs [49]. Other mutations, including TP53, SMAD4, CTNNB1, and the mammalian target of rapamycin (mTOR), are associated with advanced neoplasia and pancreatic ductal adenocarcinoma arising from mucinous cysts [49]. On the other hand, mutations in the von Hippel–Lindau (VHL) gene are associated with SCNs, and mutations in the multiple endocrine neoplasia 1 (MEN1) gene or loss of heterozygosity (LOH) are associated with pancreatic neuroendocrine tumors [49]. A recent prospective study based on 251 patients with surgical pathology was able to identify mucinous cysts with 90% sensitivity and 100% specificity via the presence of MAPK/GNAS mutations. Furthermore, the combination of MAPK/GNAS and TP53/SMAD4/CTNNB1/mTOR alterations was able to identify advanced neoplasia with 88% sensitivity and 98% specificity [49]. In this cohort of patients, the sensitivities and specificities of VHL and MEN1/LOH alterations were 71% and 100% for SCNs and 68% and 98% for pancreatic neuroendocrine tumors, respectively. Although these NGS panels are often institution-specific or send-out testing, they are becoming more commonplace in centers and provide important diagnostic information that allows clinicians to reach a more holistic decision about follow-up modalities in patients with PCLs. A summary of EUS-fluid acquisition markers and DNA mutations can be found in Table 2.

## 5. EUS-Tissue Acquisition

Unfortunately, aspirating fluid from PCLs for cytology has a low diagnostic yield for cells. As a result, the purpose of FNA extends to other ancillary tests that include analysis of cyst fluid for CEA, glucose, and DNA. Furthermore, the indications for FNA under EUS guidance are not strictly the same indications for EUS evaluation of these lesions.

The guidelines are not clear on when to use FNA for PCLs during EUS evaluation. The American Society of Gastrointestinal Endoscopy recommended in a 2016 publication to perform EUS-tissue acquisition (EUS-TA) in cystic lesions > 3 cm or in the presence of mural nodules, MPD dilation, or associated mass (the above recommendation was described as moderate quality of evidence). Conversely, the ASGE stated in asymptomatic patients with cysts < 3 cm and without mural nodules, associated mass, or MPD dilation that EUS-TA is optional (low quality of evidence) [50]. In clinical practice, many endosonographers avoid EUS-TA in the latter group when a PCL is less than 1.5–2 cm.

### 5.1. EUS-Fine Needle Aspiration (FNA)

The indications for EUS-FNA are outlined in Table 3. EUS-FNA for definite cytopathologic diagnosis has a low sensitivity of 54% (95% CI: 49–59%); however, when positive, it is highly specific at 93% (95% CI: 90–95%) [51]. Targeted cyst wall puncture after aspiration of cyst fluid was shown to improve the yield and provide an adequate specimen for cytologic interpretation in 77% of 107 cases [52]. The yield is higher at 78% of 132 cases when a mural nodule is found and sampled [53]. Typical findings when cytopathology is positive are histiocytes, macrophages, or neutrophils in a pseudocyst, glycogen-rich cuboidal cells in SCNs, and mucinous epithelial cells or extracellular mucin in mucinous lesions such as IPMNs or MCNs. Despite its low sensitivity, when TA in PCLs is indicated, EUS-FNA is the most commonly used TA modality, as it is a safe procedure with a very low risk of complications. A meta-analysis showed an overall morbidity of 2.66% (95% CI: 1.84–3.62%) and mortality of 0.19% (95% CI: 0.09–0.32%) [54]. The reported adverse events included pancreatitis (0.92%, 95% CI: 0.63–1.28%), hemorrhage (0.69%, 95% CI: 0.42–1.02%), pain (0.49%, 95% CI: 0.27–0.79%), infection (0.44%, 95% CI: 0.27–0.66%), desaturation (0.23%, 95% CI: 0.12–0.38%), and perforation (0.21%, 95% CI: 0.11–0.36%). In the case of sampling via cyst wall puncture, the rate of pancreatitis was relatively higher (2.8%) when compared to historic data using cyst fluid aspiration only [52].

### 5.2. Other EUS-TA Modalities

To combat the problems of low cellularity and poor diagnostic yield of cyst fluid analysis, researchers have developed new techniques to assess PCLs at the cellular level. The use of an EUS needle cytology brush that can be introduced through a 19-gauge needle has been reported in different studies [7,55,56]. Unfortunately, a randomized controlled multicenter trial examining the needle cytology brush did not improve the cytology yield over EUS-FNA [57]. Due to these results, in combination with difficulty in advancing the brush through the 19-gauge needle, interest in its use has faded. A more promising method of analysis of PCLs is EUS-through-the-needle biopsy (EUS-TTNB). In this technique, a microforceps device (a forceps < 1 mm in diameter) is advanced through a 19-gauge EUS needle. Using this method, clinicians biopsy the cyst wall and/or intracystic solid structures, thereby increasing cellular yield and allowing for histopathological analysis by preserving native cellular architecture. A systematic review evaluating the performance of microforceps in TA of PCLs showed an increased diagnostic accuracy of 68.6% (95% CI: 61–76%); however, this was at the expense of up to 10% adverse events, which included mild acute pancreatitis (3–7% of cases) and mostly self-limiting intracystic hemorrhage [58]. A recent prospective single-center study of 101 consecutive patients undergoing EUS-TTNB showed a complication rate of 9.9% (nine episodes of acute pancreatitis and one death) [59]; however, in this study, the diagnostic information from the biopsy led to a management change in 11.9% of cases (*n* = 12). A more recent pooled analysis reported the diagnostic yield of EUS-TTNB to be 74% and a diagnostic performance of 80%, with an adverse event rate of 5% [47]. An evaluation of EUS-TA modalities can be found in Table 4.

#### EUS and Artificial Intelligence

As EUS provides an image consisting of pixels, artificial intelligence algorithms are able to analyze pixel density and organization in order to differentiate between and risk-stratify PCLs. This is an emerging field that uses EUS images as inputs to be analyzed by neural networks in order to provide output data (i.e., mucinous vs. non-mucinous). A recent pilot study utilized 5505 images from 28 PCLs and correctly identified mucinous vs. non-mucinous PCLs with 98.3% sensitivity, 98.9% specificity, and 98.5% accuracy [60]. Additionally, deep learning is able to differentiate between high- and low-grade IPMNs by analyzing EUS images with an accuracy of 99.6%, as shown in a 2022 study analyzing 3355 EUS images from 43 patients who underwent pancreatectomy [61]. Artificial intelligence is also being applied to EUS-nCLE images and is showing similar promise [62,63]. The incorporation of real-time in vivo AI evaluation of PCLs could enable a more accurate diagnosis and risk stratification of PCLs.

## 6. Conclusions

In conjunction with cross-sectional imaging (MRI), EUS morphology is utilized for standard of care evaluation of pancreatic cysts. EUS imaging provides a more accurate morphological evaluation of intracystic features, such as wall thickness and mural nodularity. In the absence of established guidelines to dictate its use, adjunctive imaging technologies, such as EUS-nCLE and CH-EUS, can be utilized to improve management. Fluid aspiration and analysis, when performed based on indications, provide additional information. Continued research in this area should consider EUS-guided 3D imaging to augment morphological evaluation, real-time AI-guided assessment utilizing EUS morphology and EUS-guided nCLE imaging, and highly accurate cyst fluid analytics with genomics, proteomics, and metabolomics.

## Figures and Tables

**Figure 1 diagnostics-13-00705-f001:**
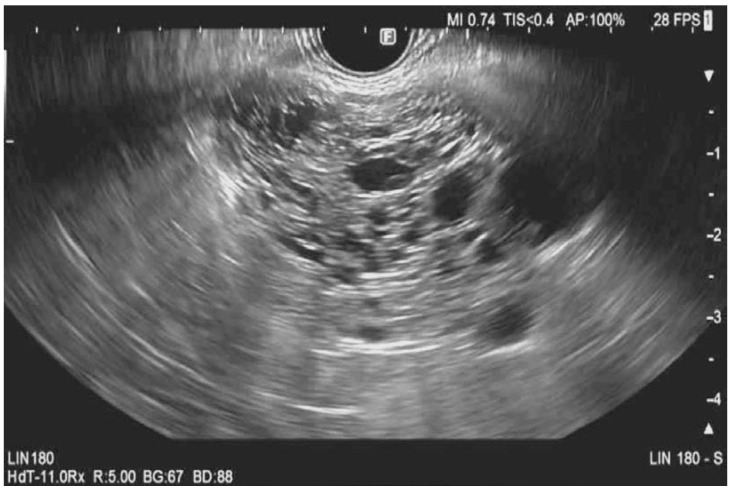
Endoscopic ultrasound morphology of a microcystic serous cystadenoma. This ultrasound image shows a microcystic serous cystadenoma with innumerable small cystic spaces separated by thin septations.

**Figure 2 diagnostics-13-00705-f002:**
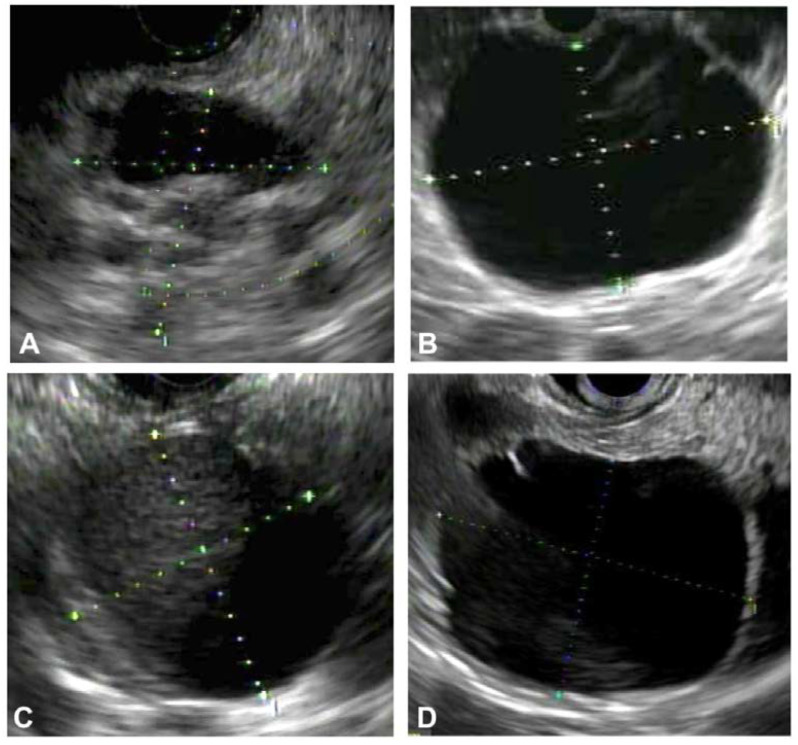
EUS appearance of various pancreatic cystic lesions. (**A**) 2.5 cm cyst with mural nodularity (11 mm); diagnosis: intraductal papillary mucinous neoplasm with high-grade dysplasia. (**B**) 6 cm thin-walled cyst; diagnosis: mucinous neoplasm with low-grade dysplasia. (**C**) 3.2 cm cyst with intracystic debris and wall thickness; diagnosis: pseudocyst. (**D**) 4.6 cm unilocular cyst; diagnosis: serous cystadenoma.

**Figure 3 diagnostics-13-00705-f003:**
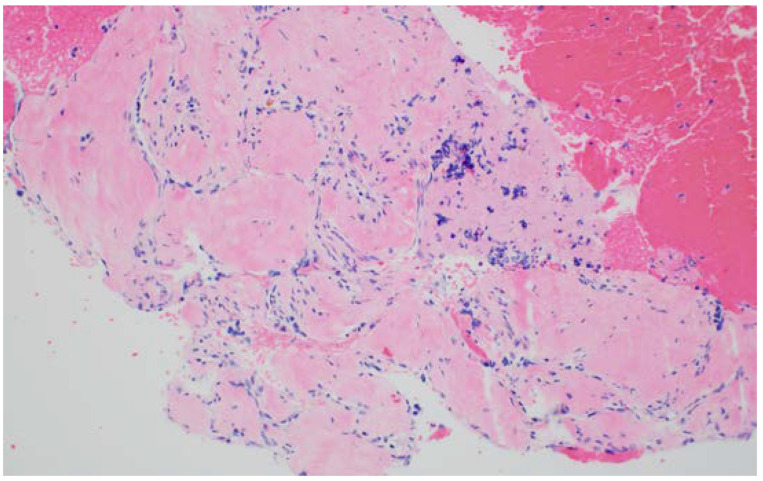
Serous cystadenoma cytopathology (courtesy of Dr. Michael G. Ozawa, Stanford University). Serous cystadenoma cytopathology showing cystic spaces separated by fibrous septa coalesced into a central scar. Hematoxylin and eosin stain, 100× magnification.

**Figure 4 diagnostics-13-00705-f004:**
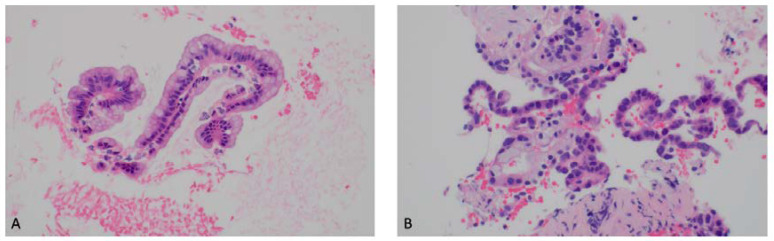
Intraductal papillary mucinous neoplasm (IPMN) histopathology (courtesy of Dr. Michael G. Ozawa, Stanford University). (**A**) Low-grade IPMN showing epithelial atypia. (**B**) High-grade IPMN showing severe epithelial atypia with necrotic debris. Hematoxylin and eosin stain, 100× magnification.

**Table 1 diagnostics-13-00705-t001:** Indications for Endoscopic Ultrasound (EUS).

Guideline	Indications
American Gastroenterological Association (AGA) (2015)	>2 high risk features: • PCL Size > 3 cm • Dilated main pancreatic duct • Presence of a solid component
International Consensus Guidelines (2017)	With any of the below features: • PCL size > 3 cm • Thickened/enhanced PCL wall • MPD 5–9 mm • Abrupt change in MPD with distal pancreatic atrophy • Lymphadenopathy • Elevated CA 19-9 • Rapid growth (>5 mm/2 years)
American College of Gastroenterology (2018)	With any of the below features: • MPD > 5 mm • IPMN or MCN > 3 cm • Change in MPD caliber with upstream atrophy • Size increase > 3 mm/year • Jaundice secondary to PCL • Pancreatitis secondary to PCL • Presence of mural nodule or solid component
European (2018)	Radiologic or clinical features of concern for malignancy: Radiologic: • MPD ≥ 5mm • Size increase ≥ 5 mm/year • Presence of mural nodule or solid component Clinical: • Jaundice secondary to PCL • New onset diabetes • Increased CA 19-9

PCL: pancreatic cystic lesion; MPD: main pancreatic duct; CA 19-9: carbohydrate antigen 19-9; IPMN: intraductal papillary mucinous neoplasm; MCN: mucinous cystic neoplasm.

**Table 2 diagnostics-13-00705-t002:** Endoscopic Ultrasound Fluid Acquisition.

**Assay**	**Pancreatic Lesion Association**
Carcinoembryonic Antigen (CEA)	Used to differentiate between mucinous and non-mucinous pancreatic cysts with varying accuracy depending on cutoff values. Generally agreed upon that CEA < 5 ng/mL is highly specific for non-mucinous PCL, with a specificity reaching 95%, and CEA > 800 ng/mL is highly specific for mucinous PCL, with a specificity of 98%.
Cyst Fluid Glucose	Newer PCL fluid marker–cyst fluid glucose of <50 mg/dL indicates a mucinous cyst with a sensitivity of 91% and specificity of 75% vs. 67% and 80%, respectively, for CEA in the same study population.
**DNA Markers**	**Pancreatic Lesion Association**
VHL Gene Mutations	von Hippel–Lindau (VHL) mutations are correlated with serous cystadenomas. This loss of function mutation has been shown to have a sensitivity of 71% and specificity of 100% in determining serous cystadenomas when evaluating pancreatic cystic lesions.
MEN1, LOH Gene Mutations	Multiple endocrine neoplasia 1 (MEN1) and loss of heterozygosity (LOH) genes are correlated with pancreatic neuroendocrine tumors. When evaluated together, these mutations have a sensitivity and specificity of 68% and 98%, respectively, in identifying a lesion as a pancreatic neuroendocrine tumor.
MAPK, GNAS Mutations	MAPK and GNAS mutations are used to identify a pancreatic cystic lesion as mucinous with a sensitivity of 90% and specificity of 100%.
TP53, SMAD4, CTNNB1, mTOR	When combined with MAPK/GNAS mutation in a sequencing panel, the addition of TP53/SMAD4/CTNNB1/mTOR was able to identify advanced neoplasia with a sensitivity and specificity of 88% and 98%, respectively.

CEA: carcinoembryonic antigen; VHL: von Hippel–Lindau; MEN1: multiple endocrine neoplasia 1; LOH: loss of heterozygosity; MAPK: mitogen-associated protein kinase; GNAS: guanine nucleotide binding protein–alpha subunit; TP53: tumor protein p53; mTOR: mammalian target of rapamycin.

**Table 3 diagnostics-13-00705-t003:** Indications for Endoscopic Ultrasound Cyst Fluid Acquisition.

Guideline	Indications
American Gastroenterological Association (AGA) (2015)	EUS-FNA if >2 high risk features: • Cyst size > 3 cm • Dilated main pancreatic duct • Presence of a solid component
International Consensus Guidelines (2017)	EUS-FNA if PCL with any of the below worrisome features: • Cyst size > 3 cm • Thickened/enhanced cyst walls • MPD 5–9 mm • Abrupt change in MPD with distal pancreatic atrophy • Lymphadenopathy • Elevated CA 19-9 • Rapid growth (>5 mm/2 years)
American College of Gastroenterology (2018)	EUS-FNA if PCL with any of the below features: • MPD > 5 mm • IPMN or MCN > 3 cm • Change in MPD caliber with upstream atrophy • Size increase > 3 mm/year • Jaundice secondary to cyst • Pancreatitis secondary to cyst • Presence of mural nodule or solid component
European (2018)	For cysts when the diagnosis is unclear and the results are expected to change clinical management. EUS-FNA is not to be performed if the diagnosis is available or if there is a clear indication for surgery.
American Society for Gastrointestinal Endoscopy	EUS-FNA recommended for PCLs > 3 cm, presence of an epithelial nodule, dilated MPD, or suspicious mass lesion. In the absence of these features, EUS-FNA is considered optional.

EUS-FNA: endoscopic ultrasound fine needle aspiration; MPD: main pancreatic duct; CA 19-9: carbohydrate antigen 19-9; IPMN: intraductal papillary mucinous neoplasm; MCN: mucinous cystic neoplasm.

**Table 4 diagnostics-13-00705-t004:** Endoscopic Ultrasound Tissue Acquisition Modalities.

Technique	Description
Endoscopic Ultrasound- Fine Needle Aspiration (EUS-FNA)	Low sensitivity of 54%, but when positive, has a specificity of 93%. Targeted cyst wall puncture after aspiration of cyst fluid was shown to improve the yield and provide an adequate specimen in 77% of cases.
Endoscopic Ultrasound Throught-the-Needle Biopsy (EUS-TTNB)	Introduction of microforceps through a 19-gauge needle, allowing for maintainence of cellular structure and potentially a higher yield. Pooled analysis showed a diagnostic yield of 74% and a diagnostic performance of 80%.

## Data Availability

Review article did not require data analysis.

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
