# Peer review of "Role of Endoscopic Ultrasound in the Evaluation of Pancreatic Cystic Neoplasms: A Concise Review"

_diagnostics, 2023, doi:10.3390/diagnostics13040705_

Round 1

Reviewer 1 Report

Congratulations to the authors who gave a complete overview of the most common pancreatic lesions that could be identified by US.

EUS morphology is utilized for standard of care evaluation of pancreatic cysts In the work the authors described in a very detailed manner the association between radiological findings and clinical features of pancreatic cystic lesions.

We believe that the paper is suitable for publication after minor revisions

Author Response

Reviewer 1:

Congratulations to the authors who gave a complete overview of the most common pancreatic lesions that could be identified by US.

EUS morphology is utilized for standard of care evaluation of pancreatic cysts In the work the authors described in a very detailed manner the association between radiological findings and clinical features of pancreatic cystic lesions.

Thank you for the time you took to review our manuscript as well as the feedback. We appreciate your assistance throughout the process and we are honored to educate your readership on the evaluation of pancreatic cysts.

We believe that the paper is suitable for publication after minor revisions

Reviewer 2 Report

Dear Authors,

I have read with interest the present brief review on the EUS management of pancreatic cystic neoplasms.

The article is well-written and organized. 

Minor changes:

- I should include also a table describing the outcomes of biochemical analysis of cystic fluid aspirate

- Another table on the outcomes of EUS-guided tissue acquisition outcomes.

- I should change the acronyms pancreatic cystic lesion (PCL) into pancreatic cystic neoplasm (PCN)

Author Response

Reviewer 2:

Dear Authors,

I have read with interest the present brief review on the EUS management of pancreatic cystic neoplasms.

The article is well-written and organized. 

Minor changes:

- I should include also a table describing the outcomes of biochemical analysis of cystic fluid aspirate

Thank you for the time and diligence taken to review our manuscript. We believe your edits and recommendations strengthen the paper as a whole and provide more clarity and information to your readership. We have included a Table describing the outcomes of biochemical analysis of cyst fluid aspirate. This is now called Table 2. The previous Table 2 in the originally sent manuscript is now renamed as Table 3.

- Another table on the outcomes of EUS-guided tissue acquisition outcomes.

We are in agreement that a cleaner side-by-side evaluation of EUS-FNA vs. EUS-TTNB will give the readership more clarity, therefore we have included this in Table 4. We hope this concise table gives a clearer definition and performance assessment of these EUS tissue acquisition modalities.

- I should change the acronyms pancreatic cystic lesion (PCL) into pancreatic cystic neoplasm (PCN)

We believe that the term pancreatic cystic lesion (PCL) is more appropriate that pancreatic cystic neoplasm. This is because a lesion is a PCL until proven to be neoplastic with diagnostic modalities – only then does it become a pancreatic cystic neoplasm. Not all of the lesions discussed in this paper are pancreatic cystic neoplasms. Therefore, we choose to keep the term PCL in lieu of changing the term in the body of the paper to pancreatic cystic neoplasm. Thank you.